# COVID-19 Vaccination Campaign among the Health Workers of Fondazione Policlinico Universitario Agostino Gemelli IRCCS: A Cost–Benefit Analysis

**DOI:** 10.3390/ijerph19137848

**Published:** 2022-06-26

**Authors:** Mario Cesare Nurchis, Alberto Lontano, Domenico Pascucci, Martina Sapienza, Eleonora Marziali, Francesco Castrini, Rosaria Messina, Luca Regazzi, Francesco Andrea Causio, Andrea Di Pilla, Giuseppe Vetrugno, Gianfranco Damiani, Patrizia Laurenti

**Affiliations:** 1Department of Woman and Child Health and Public Health, Fondazione Policlinico Universitario A. Gemelli IRCCS, 00168 Rome, Italy; mariocesare.nurchis@unicatt.it (M.C.N.); giuseppe.vetrugno@unicatt.it (G.V.); gianfranco.damiani@unicatt.it (G.D.); patrizia.laurenti@unicatt.it (P.L.); 2Department of Health Sciences and Public Health, Section of Hygiene, Catholic University of the Sacred Heart, 00168 Rome, Italy; domenico.pascucci@outlook.it (D.P.); martina.sapienza01@icatt.it (M.S.); eleonora.marziali01@icatt.it (E.M.); francesco.castrini01@icatt.it (F.C.); rosaria.messina01@icatt.it (R.M.); luca.regazzi01@icatt.it (L.R.); francescoandrea.causio01@icatt.it (F.A.C.); andrea.dipilla01@icatt.it (A.D.P.); 3Department of Health Care Surveillance and Bioethics, Section of Legal Medicine, Catholic University of the Sacred Heart, 00168 Rome, Italy

**Keywords:** economic evaluation, COVID-19, vaccine

## Abstract

Vaccinations generate health, economic and social benefits in both vaccinated and unvaccinated populations. The aim of this study was to conduct a cost–benefit analysis to estimate the costs and benefits associated with the COVID-19 vaccination campaign for health workers in Fondazione Policlinico Universitario Agostino Gemelli IRCCS (FPG). The analysis included 5152 healthcare workers who voluntarily received the Pfizer–BioNTech COVID-19 vaccine, divided into physicians, nurses and other health workers. Data about vaccine cost, administration and materials were derived from administrative databases of the FPG from 28 December 2020 to 31 March 2021. The costs associated with the COVID-19 vaccination campaign amounted to EUR 2,221,768, while the benefits equaled EUR 10,345,847. The benefit-to-cost ratio resulted in EUR 4.66, while the societal return on investment showed a ratio of EUR 3.66. The COVID-19 vaccination campaign for health workers in FPG has high social returns and it strengthens the need to inform and update decision-making about the economic and social benefits associated with a vaccination campaign. Health economic evaluations on vaccines should always be considered by decision-makers when considering the inclusion of a new vaccine into the national program.

## 1. Introduction

Burden of disease assessment estimates the impact of a disease on a population by measuring the morbidity and mortality attribution and presenting it through a metric called disability-adjusted life years (DALYs) [1]: this measure considers years of life lost (YLLs) and years lost due to disability (YLDs).

The DALY method can be also adopted as an outcome measure when conducting cost-effectiveness analyses [2], which are of paramount importance in the decision-making process led by policy makers; indeed, DALYs can be used to compare health in two populations, equate health conditions within the same population in different times and provide information to prioritize and allocate healthcare resources [3].

DALYs also represent a tool to determine the effects of mass vaccination, which constitutes a fundamental component of primary healthcare and is responsible for improved health outcomes globally [4,5].

In low- and middle-income countries, vaccinations against 10 diseases have prevented 37 million deaths between 2000 and 2019, of which 36 million were deaths averted in children under the age of 5 years [6].

Vaccinations generate both health and non-health benefits in both vaccinated and unvaccinated populations, including indirect benefits and spillover effects [7]. The benefits on health are the most significant and intuitive ones: vaccination campaigns lead to tremendous decreases in infectious diseases, morbidity and mortality, and this effect is even more accentuated in low- and middle-income countries [5,8]. Not only do vaccines grant personal, direct protection, but they also have the potential to generate herd immunity, representing a great resource for the whole population, particularly for those too young, too vulnerable or too immunosuppressed to receive them. Furthermore, vaccines can prevent secondary infections that complicate vaccine-preventable diseases, such as bacterial infections, and non-communicable diseases, such as cancer. Another important effect of vaccination campaigns is the prevention of antibiotic resistance as they contrast the development of viral and bacterial infections and reduce the antibiotic burden to which the microbiota of patients are exposed [9,10].

In the face of investments made by governments to purchase vaccines and administer infrastructure, a reduced incidence of diseases and their associated treatments and healthcare costs follows [9]: this can lead to economic growth and less absenteeism at work for patients and their caregivers [11], particularly for women [12], who are usually in charge of children and elderly people [5].

Promoting vaccinations is also an investment in human capital since it can lead to improved educational attainment for children [13,14,15], fewer child impairments and, as a result, a larger and more productive adult workforce [14,16]; the benefits of vaccination can extend even further, as economic growth can determine lower fertility rates [10,17], improved life expectancy and a widespread attitude of investing in the education of children and personal savings [18].

Additionally, vaccination campaigns can have social benefits: they can level the impact of infectious diseases across all social strata, reducing inequities [5,19]; moreover, infrastructure needs to be implemented to provide vaccinations universally: this can foster the spread of other health services among fragile social categories such as children, pregnant women and elderly people [20]. Finally, already available public health infrastructure can be leveraged for health promotion [20].

Coronavirus disease 2019 (COVID-19) has emerged as one of the most appalling threats to world health in recent times.

As of October 2021, more than 250 million cases have been diagnosed and over 5 million deaths have occurred globally [21]. Only with regard to Italy, almost 5 million people were infected, among which 146,000 were healthcare workers (HCWs) [22].

Concerning the burden related to COVID-19, it seems to be owed mainly to mortality, whereas the contribution of the disability weight is significantly lower [23,24]: consequently, health policies should prioritize issues related to COVID-19 fatality [25].

Traditionally, economic evaluations of vaccines combine information about the incidence of the disease to be prevented, the probability of sequelae, the clinical effectiveness of the vaccine, its impact on productivity loss and the costs incurred, which may include those for treatment of the disease and its consequences, administration of the vaccine and treatment of potential adverse effects [26,27,28].

However, this is not an all-round approach, as it neglects the abovementioned effects on health, economy and society.

The implications of proper vaccine evaluations are not merely academic [26], and health economic evaluations (HEEs) on vaccines and vaccination programs should always be considered by decision-making bodies when considering the inclusion of a new vaccine into the national program or negotiating prices with manufacturers [29]. How economic evaluations affect institutional decisions, however, depends on the context of the evaluation and whether resources must be allocated within a fixed budget [27].

Therefore, the aim of this study is to conduct a cost–benefit analysis to estimate the costs and benefits associated with the COVID-19 vaccination campaign for HCWs at Fondazione Policlinico Universitario Agostino Gemelli IRCCS.

## 2. Materials and Methods

The pharmacoeconomic evaluation was designed as a cost–benefit analysis (CBA) to assess the net economic benefit of the severe acute respiratory syndrome coronavirus 2 (SARS-CoV-2) vaccination program among HCWs in a teaching hospital. The null hypothesis was that the benefits of a COVID-19 vaccination campaign among HCWs were lower than the related costs.

### 2.1. Setting and Population

The study took place in the Fondazione Policlinico Universitario Agostino Gemelli IRCCS (FPG). The study population consisted of 5152 HCWs who voluntarily received the Pfizer–BioNTech COVID-19 vaccine, divided into 2415 physicians, 1717 nurses and 1020 other HCWs. The analysis considered only one type of COVID-19 vaccine (i.e., BNT162b2) due to regulations at regional level.

### 2.2. Costs and Benefits Estimation

The analysis of the vaccination program included both direct costs, including the costs of vaccines and materials and the cost of administration by personnel, and indirect costs, including the average time lost for vaccine administration and the cost of work absenteeism due to vaccine side effects.

The cost of vaccine and materials was established according to their purchasing prices while the cost of personnel involved in the administration of vaccine was computed considering the main tasks essential to the delivery of the Pfizer–BioNTech COVID-19 vaccine and quantifying the operative time of the personnel involved (i.e., physicians, nurses, pharmacists, health assistants, administrative staff, security guards).

The cost of working time lost for vaccination was calculated by taking into account the time spent by HCWs out of work for vaccination and their mean hourly wage; the cost of work absenteeism due to adverse events was estimated by considering the number of working days lost due to COVID-19 (i.e., 14 days [30]) and the mean daily wage.

The benefits directly and indirectly associated with the vaccination campaign included the costs avoided due to reduced hospitalizations and the contained costs of temporary and permanent absenteeism from work (i.e., productivity loss).

The cost averted due to fewer hospitalizations was estimated considering COVID-19 hospitalization rate [31], its relative cost [32] and vaccine efficacy. The temporary and permanent productivity loss averted was determined according to the human capital approach (HCA) [33,34,35] methodology.

Temporary productivity loss (TPL) was evaluated based on the number of HCWs, days away from work due to COVID-19 [36] and the daily mean wage stratified by job category [37], all retrieved from selected sources.

The TPL for HCWs in each job category (i.e., physicians, nurses, and other HCWs), was calculated as the daily mean wage of the HCWs in each job category, adjusted by the vaccine effectiveness, by the COVID-19 infection rate among HCWs and by the number of days absent from work due to COVID-19. When the time of absence from work was less than 12 months, the discount was not applied. Then, the total TPL was the sum of the TPL of each job category.

For permanent productivity loss (PPL), the case fatality rate among HCWs [38], vaccine efficacy [39], average age at death among HCWs [37,38], retirement age [40] and the annual wage stratified by job category [37] were obtained from the selected sources.

Estimates of PPL for a specific job category Wj were computed as the sum of the discounted annual mean wages of the HCWs in each job category, for each PYLLj, multiplied by the number of HCWs in each job category, Wj, adjusted by the vaccine efficacy and the case fatality rate among HCWs.
$$\mathrm{PPL}=\overset{PYLL_{j}}{\underset{t=0}{\sum }}\frac{{\mathrm{annual}\;\mathrm{mean}\;\mathrm{wage}}_{j}}{{\left(1+r\right)}^{2}}\times \left(\mathrm{adjusted}\;\mathrm{number}\;\mathrm{of}\;\mathrm{HCWs}\;\mathrm{in}\;\mathrm{each}\;\mathrm{job}\;\mathrm{category}\right).\;$$

Then, the total PPL was the sum of the PPL of each job category.

The PYLLs were calculated as the difference between the retirement age and the average age at death due to COVID-19 among HCWs. The retirement age was set at 67 [39], as established by the Istituto Nazionale della Previdenza Sociale (INPS), while the average age at death due to COVID-19 among HCWs was 52, as reported by the Istituto Superiore di Sanità (ISS) [37]. A 3% discount rate was adopted to discount the annual mean wages for each job category.

The implementation of a healthcare program could be seen as an investment in individual human capital. In assessing the return of healthcare investments, the value of healthy time gained could be evaluated in terms of individual increased production in the marketplace [41].

### 2.3. Cost Benefit Analysis

A cost–benefit analysis [42,43] of implementing COVID-19 vaccination in a teaching hospital was conducted from the healthcare system perspective. The time horizon was approximately 3 months. In light of the abovementioned timespan, the discounting was not applied.

The benefit-to-cost ratio (BCR) is the most widely used indicator in CBA showing the relationship between the relative costs and benefits of a proposed program. The BCR was computed as follows:$$\mathrm{BCR}\;=\frac{\mathrm{Total}\;\mathrm{benefits}}{\mathrm{Total}\;\mathrm{costs}}$$

The societal return on investment (SROI) represents a powerful metric to understand the potential financial benefits of an investment from the perspective of a healthcare system. The SROI was computed according to the following formula [44,45]:$$\mathrm{SROI}=\frac{\left(\mathrm{Total}\;\mathrm{benefits}-\mathrm{Total}\;\mathrm{costs}\right)}{\mathrm{Total}\;\mathrm{costs}}.\;$$

In order to allow and facilitate the decision-making process, the decision rule is that if these ratios are greater than 1, then the program produces more benefits than costs.

In addition to the SROI and BCR, the net benefits (NB) were estimated, as follows, to provide an absolute measure of benefits:NB=Total benefits −Total costs

Therefore, programs with positive NB (i.e., NB > 0) are considered to be viable.

### 2.4. Sensitivity Analysis

The quantified costs and benefits of a program may vary based on several assumptions on input data and methodology applied in the conducting of the cost–benefit analysis. Thus, sensitivity analysis was performed to deal with uncertainties in the input variables and to gauge the range of potential SROIs. A univariate deterministic sensitivity analysis was run to assess the effect of changing input parameters one by one (i.e., personnel, materials, sick leave for COVID-19, COVID-19 infection rate among HCWs) while holding the remaining values unvaried.

## 3. Results

The socio-demographic characteristics of the analyzed sample have been described elsewhere [39]. The cost of materials amounted to EUR 126,453 while the cost of vaccine administration accounted for EUR 138,017. Furthermore, Table 1 showed the main cost and benefit items stratified by the three job categories.

Overall, the costs associated with the COVID-19 vaccination campaign in the teaching hospital amounted to EUR 2,221,768, of which 85% was related to the costs for work absenteeism due to vaccine side effects. Benefits equaled EUR 10,345,847, of which 98.5% was related to the PPL averted.

For what concerns the NB, the absolute measure accounted for EUR 8,124,079, thus, sharply greater than zero.

The BCR resulted in EUR 4.66, implying that the present benefits exceed the present value of the costs. Particularly, for every EUR 1 the vaccination campaign costs, there would be EUR 4.66 in benefits.

In the base case scenario, the calculation of the SROI showed a ratio of EUR 3.66, highlighting that investing EUR 1 in the COVID-19 vaccination campaign would generate a social return of EUR 3.66.

Furthermore, univariate deterministic sensitivity analyses were conducted by varying the personnel cost for vaccine administration and the consumables cost by 25%, doubling and halving the COVID-19 infection rate among HCWs, and multiplying and dividing by three the number of sick leaves for COVID-19. Figure 1 illustrates the findings of the univariate deterministic sensitivity analyses. The base case SROI is mainly sensitive to changes in the number of sick leaves for COVID-19 and consumables and personnel costs, whereas it is less sensitive to changes in COVID-19 infection rate among HCWs.

Given these results, the number of sick leaves for COVID-19 had the greatest impact on base case SROI, increasing it up to 3.79. Additionally, a 25% reduction in the cost of personnel to administer the vaccine and the cost of consumables raised the SROI to 3.74, while an increase of 25% decreased the SROI to 3.59. Additionally, with respect to the base case, assuming a doubling of the COVID-19 infection rate, the SROI increased up to 3.74, while supposing a halving of the same rate, the SROI was lowered to 3.61.

## 4. Discussion

The study results pointed out that benefits associated with the COVID-19 vaccination campaign in FPG sharply exceed the relative costs. Heterogeneous values of costs and benefits were observed across the three job categories considered, with the highest amounts among physicians, followed by nurses and other HCWs, mainly due to the differences in salaries among these groups.

Findings from the cost–benefit analysis were explored by a univariate deterministic analysis highlighting that implementing the COVID-19 vaccination campaign would generate a social return in all the assessed scenarios, as already shown for the flu vaccination [46].

Spurred by the widespread diffusion of the COVID-19 pandemic, the development of vaccines has been accelerated, with four vaccines currently officially administered in Italy. Thus, along with the gradual shortening of the gap between demand and supply, it is urgent for economic appraisals to steer an informed decision-making process [47,48].

In line with the current general agreement, our results corroborate the strategy of vaccinating prioritized categories such as HCWs as well as the general population. HCWs absenteeism due to SARS-CoV-2 infection has been a burdensome problem in the COVID-19 pandemic since they represent the vital workforce facing this pandemic. Results from a recent study showed that, among 952 HCWs who tested positive for COVID-19 between December 2020 and July 2021, return-to-work time for fully vaccinated HCWs (10.9 days) was significantly shorter than that of partially vaccinated HCWs (15.5 days), which in turn was significantly shorter than that of unvaccinated HCWs (18.0 days). Fully vaccinated HCWs also showed milder symptom profiles compared to partially vaccinated and unvaccinated HCWs. Given the above results, COVID-19 vaccination has the potential to prevent long absences from work and the adverse financial, staffing and managerial consequences of these long absences [49].

The above implication is also supported by recent evidence in the scientific literature, even though it is not specifically focused on HCWs vaccination. Lopez et al. conducted a cost–benefit analysis of the vaccination campaign in Spain and reported that mass vaccination is cost-saving [50]. Following the same methodology, Wang et al. outlined that all the investigated vaccines dominated the no-vaccination strategy, reporting high BCRs [51].

The study by Pearson et al. found that a one-year COVID-19 vaccination campaign is likely to be cost-effective in several scenarios [52]. Moreover, Padula et al. showed a probability of 70% for mass vaccination strategy to be cost-effective at a given willingness-to-pay threshold (i.e., USD 50,000).

The combination of study results paves the way for another main implication. Given the present situation, characterized by the uncertainty due to new lineages (e.g., B.1.1.529) and a steady trend of disease incidence, it is of paramount importance to encourage critical thinking about how to develop and implement immunization health policies at all levels of decision-making. Hence, decision-makers and all the stakeholders involved in the process should orient their attention to an investment perspective for the National Health System, being cautious to avoid any possible containment actions owing to resource constraints and promoting communication strategies to inform and convince individuals, even the most reluctant ones, of the clinical–epidemiological, economic and social benefits deriving from vaccination.

The findings of the study must be considered in light of their weaknesses and strengths. First of all, the CBA analysis did not compute incremental costs and benefits related to the vaccination campaign. However, this may be explained by the rapid evolution of the pandemic; indeed, the time horizon was set at 3 months, which roughly corresponds to the time lag between vaccination and the possibility of reinfection by SARS-CoV-2 [53]. In addition, the analysis was focused only on the vaccination strategy, not including other levels of health and containment measures. Nevertheless, following a robust methodology widely referenced in literature [42], we tried to encompass all the possible direct and indirect cost and benefit items related to the vaccination campaign. Additionally, the cost per dose of vaccine was assumed as international and national bodies did not provide any official data. Furthermore, a convenience sample in only one research hospital was considered for the analysis. Nonetheless, the analyzed sample of HCWs was quite large, allowing us to conduct the economic assessment. A further limit was the lack of detailed data about the shifts of HCWs in COVID-19-dedicated wards, which did not allow for distinction of patient-facing and nonpatient-facing roles, with possible alterations in the COVID-19 infection rate. An additional limit was the lack of consideration of the mean absenteeism before COVID-19 in the analysis. Moreover, another caveat was the lack of consideration of whether and to what extent the effectiveness of the vaccine would be conditioned by the newest variants of concern, such as the B.1.359 and the B.1.1.529 variants.

Further economic evaluations are needed to assess the implementation of COVID-19 vaccination strategies among HCWs and the general population, considering also the effect of new lineages on the effectiveness of the vaccine. Additional research is required to investigate the cost-effectiveness, and not only the cost savings of COVID-19 vaccination strategies to scrutinize a key variable such as the durability of immunity. Furthermore, future economic evaluations, sharing the same methodological assumptions, need to be conducted considering several hospitals to obtain an objective view for the whole society.

## 5. Conclusions

Our findings show huge benefits with respect to costs in implementing a COVID-19 vaccination campaign in our research hospital. In general, vaccination represents one of the greatest medical discoveries ever made by mankind and is comparable in importance, in terms of health impact, to the ability to provide fresh water to a population [54]. The analysis concludes that the COVID-19 vaccination campaign for health workers in FPG has high social returns. Lessons learned from the present study strengthen the need to inform and update decision-making about the economic and social benefits associated with a vaccination campaign in order to steer targeted choices of health policy at the macro-, meso-, and micro-level.

## Figures and Tables

**Figure 1 ijerph-19-07848-f001:**
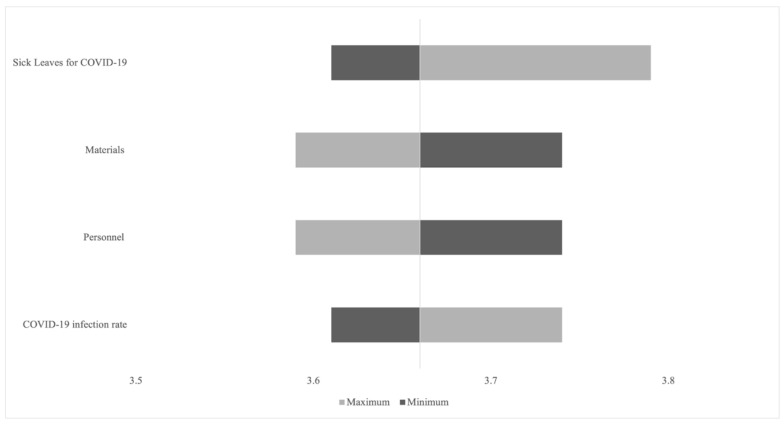
Tornado chart showing a set of univariate deterministic sensitivity analyses. The vertical line represents the base-case SRPI (EUR 3.66), while the tails of each bar show the changes in SROI when each parameter is varied.

**Table 1 ijerph-19-07848-t001:** Cost and benefits items according to the job categories.

Items	Physicians	Nurses	Other HCWs
** *Costs* **			
Cost of average time lost to be vaccinated	€41,972.7	€14,515.5	€7338.9
Cost for work absenteeism due to side effects	€1,259,084.4	€414,209.1	€220,177.2
** *Benefits* **			
Cost averted due to fewer hospitalizations	€327	€232	€138
Cost of work absenteeism averted (TPL)	€102,128.1	€33,597.7	€17,859.2
Costs of work absenteeism averted (PPL)	€6,776,502.7	€2,229,418.4	€1,185,643.8

Abbreviations: HCWs, health care workers; TPL, temporary productivity loss; PPL, permanent productivity loss.

## Data Availability

Data about vaccine cost, administration, materials, time spent by the individuals to be vaccinated, information on the number of healthcare workers and employees’ hourly wages and working hours were retrieved from the 2021 administrative databases of the teaching hospital from 28 December 2020 to 31 March 2021. The consolidated sources available from the World Health Organization (WHO) archives were used to obtain the number of sick leaves due to COVID-19 and the number of sick leaves due to adverse reactions to COVID-19 vaccination, while information on retirement age, labor force participation rate and unemployment rate was obtained from the Istituto Nazionale della Previdenza Sociale (INPS) and Istituto Nazionale di Statistica (ISTAT), respectively. The most up-to-date report, published by the Istituto Superiore di Sanità (ISS), was used to retrieve the average age at death due to COVID-19 among healthcare workers. The data presented in this study are available on request from the corresponding author.

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
