# Peer review of "COVID-19 Vaccination Campaign among the Health Workers of Fondazione Policlinico Universitario Agostino Gemelli IRCCS: A Cost–Benefit Analysis"

_ijerph, 2022, doi:10.3390/ijerph19137848_

Round 1

Reviewer 1 Report

The submitted manuscript addresses the important topic of cost-benefit analysis of covid-19 vaccination.

The manuscript is written in correct language.

The components of the manuscript are written according to generally accepted principles as for any scientific report.

The objective of the study is correctly defined.

The methodology of the study is not objectionable. 

The results are presented in a clear way. 

The discussion is comprehensive.

My comment:

- lack of explanation of the abbreviation: COVID-19; SARS-CoV-2; 

- In methodology - there should be spaces between models and text;

- not very readable - Figure 1; 

- conclusions should be formulated like points, referring directly to the results.

It is worth noting that this is a study involving a professional group of only 1 medical facility. Perhaps it is worth mentioning that for an objective view of the whole society, the study should be conducted in other medical units, of course using the same methods. 

Bibliography:

- add DOI to each publication;

- [1] in the entry:

13. Keja K, Chan C, Hayden G, Henderson RH. Expanded Programme on Immunization / Ko Keja ... [et al]. World health 339

statistics quarterly 1988 ; 41(2) : 59-63 1988 - incorrect transcript;

31. European Observatory on Health Systems and Policies. Easing of measures (transition measures): Health financing. . - no reference;

42. Pan American Health Organization. Cost-Benefit Analysis Methodology. . 2019 - no reference;

Reviewer 2 Report

First of all, congratulations on your work. It was a pleasure to read. Please find the recommendations below as an opportunity to improve the scientific soundness of your article.

Abstract

Line 17 - Change "wheconsidering" to "when considering". As in line 16, you already have "consider*" a change in the verb could be more aesthetic.

Abstract well built, representing the multiple parts of the article.

Introduction

The introduction requires an overall organization. There is no flow. As a recommendation, consider talking about DALYs -> vaccination -> COVID -> COVID vaccination -> research problem. The information is there, but it is not well organized.

Material and methods

Line 126 - Could you describe the period taken into account for the number of days lost?

Did you take into account the COVID waves or assess an overall mean?

Was mean absenteeism before COVID included in the equation? The pandemic may mask the total number of absenteeism due to other reasons. Indicating the shift from pre-COVID to COVID absenteeism could help understand the severity of the problem. Although mentioning this in this section, it could be placed in another part (Introduction or Discussion) if you feel it is an important perspective. Please do not feel obliged to integrate this recommendation, as it is only meant to help replicability.

Did you consider absenteeism due to the presence of COVID in the family and not due to infection of the worker? This is not mandatory, as it may be hard to calculate. It is just something that would be interesting to be included if it has been done.

Overall, a very well described methodology. Congratulations.

Results

A table regarding the cost and benefits estimation and analysis would be welcome as it would help the reader have an immediate perspective of the findings.

This section seems a bit short. Much of the information that should be included in the results section is in the discussion section. Merging the results and discussion sections could be helpful.

Discussion

Line 233 (please add indentation).

Very well described and interesting discussion.

Conclusions

Conclusions are congruent with the findings.

Other recommendations

Although I am not an English native, it was sometimes hard for me to read parts of the article (more specifically, in the introduction section). Consider using a language editing service or a software program (e.g. Grammarly) to improve readability. Grammarly can be used freely and will help improve the manuscript's grammar.

Reviewer 3 Report

Dear authors, thank you for the conducted study.

According to the text of the manuscript, the conducted study 'included 5,152 healthcare workers, who voluntarily received the Pfizer BioNTech COVID-19 vaccine, divided into physicians, nurses and other health workers.' It was a great idea to divide the HCWs according to their status/job.

With that, there are some concerns.

1)      The study population was divided into three groups. It is a good idea, as the results of those groups are expected to vary. It was established that the risk of exposure to coronavirus disease (COVID-19) is higher in HCW in patient-facing roles than in their colleagues in nonpatient-facing roles, although the seropositivity rate is approximately twofold higher even in HCWs in nonpatient-facing roles (as they are at increased risk exposure) than in the general population (PMID: 32870978, PMID: 32962975, PMID: 32962975). In addition, the risk of COVID-19 infection increases with age and can have gender differences and the hospitalization rate. Lastly, the salaries among those groups are also expected to be different and that should have caused the differences in the cost/benefit ratio. With that, there is no comparison in the text among the groups. Give us the information and compare the HCW groups depending on their job type.

2)      Please provide the information about sample size calculations.

3)      Please give the null hypothesis.

4)      According to the data from the literature, the Pfizer vaccine accounted for 70%, AstraZeneca for about 15% and Moderna for about 12-15% of the vaccines taken. Give information about the region described in the study, and why was the study limited to analysis of a single vaccine?

5)      Line 154. What did you mean? 'The retirement age was set at 6739, as established by the INPS while the average age at death'.

Overall, without the comparison of the HCWs from the perspective of their job type, the division is not necessary, and the manuscript lacks the novelty.

Round 2

Reviewer 3 Report

Dear authors, the conducted revision significantly improved the clarity of the text and its scientific soundness. Thank you. I can only recommend that you change the title of the manuscript. The present seems too long (COVID-19 Vaccination Campaign among the Health Workers of Fondazione Policlinico Universitario Agostino Gemelli IRCCS: A Cost-Benefit Analysis). It will probably be better to change the 'of Fondazione Policlinico Universitario Agostino Gemelli IRCCS' to 'university hospital', but it depends on yours.